# Neural Causal Structure Discovery from Interventions

**Nan Rosemary Ke**[*]                                             *rosemary.nan.ke@gmail.com*
*Google Deepmind*

**Olexa Bilaniuk**[*]                                                     *obilaniu@gmail.com*
*Mila*

**Anirudh Goyal**                                                 *anirudhgoyal9119@gmail.com*
*Google Deepmind*

**Stefan Bauer**                                                             *st.bauer@tum.de*
*Technical University of Munich*

**Hugo Larochelle**                                                   *hugolarochelle@google.com*
*Google Deepmind*

**Bernhard Schölkopf**                                                 *bs@tuebingen.mpg.de*
*Max Planck Institute for Intelligent Systems*

**Michael C. Mozer**                                                       *mcmozer@google.com*
*Google Deepmind*

**Chris Pal**                                                     *christopher.pal@polymtl.ca*
*Mila, Polytechnique Montreal*

**Yoshua Bengio**                                                 *yoshua.bengio@mila.quebec*
*Mila, University of Montreal, CIFAR Senior Fellow*

**Reviewed on OpenReview:** *https://openreview.net/forum?id=rdHVPPVuXa*

## Abstract

Recent promising results have generated a surge of interest in continuous optimization methods for causal discovery from observational data. However, there are theoretical limitations on the identifiability of underlying structures obtained solely from observational data. Interventional data, on the other hand, provides richer information about the underlying data-generating process. Nevertheless, extending and applying methods designed for observational data to include interventions is a challenging problem. To address this issue, we propose a general framework based on neural networks to develop models that incorporate both observational and interventional data. Notably, our method can handle the challenging and realistic scenario where the identity of the intervened upon variable is unknown. We evaluate our proposed approach in the context of graph recovery, both de novo and from a partially-known edge set. Our method achieves strong benchmark results on various structure learning tasks, including structure recovery of synthetic graphs as well as standard graphs from the Bayesian Network Repository.

## 1 Introduction

Structure discovery concerns itself with the recovery of the graph structure of Bayesian networks from data. When Bayesian networks are used to model cause-effect relationships and are augmented with the notion of

---
[*]Equal contributions

*interventions* and counterfactuals, they can be represented as structural causal models (SCM). While Bayesian networks can uncover statistical correlations between factors, SCMs can be used to answer higher-order questions of cause-and-effect, up in the ladder of causation (Pearl & Mackenzie, 2018). Causal structure learning using SCMs has been attempted in several disciplines including biology (Sachs et al., 2005; Hill et al., 2016), weather forecasting (Abramson et al., 1996) and medicine (Lauritzen & Spiegelhalter, 1988; Korb & Nicholson, 2010).

Structure can be learned from observational or interventional data. Observational data is sampled from the distribution without interventions; alone, it contains only limited information about the underlying causal graph (Spirtes et al., 2000). Without making restrictive assumptions about the data generating process (Shimizu et al., 2006), interventional data is needed in order to fully identify the true causal graph (Eberhardt & Scheines, 2007; Eberhardt, 2012; Eberhardt et al., 2012).

Recently, there has been a surge of interest in using continuous optimization methods, often involving neural networks, to discover causal relationships from observational data (Zheng et al., 2018; Yu et al., 2019). These methods frame the search for a causal graph or directed acyclic graph (DAG) as a continuous optimization problem, avoiding the need to search through a super-exponential number of graphs (Heinze-Deml et al., 2018a). While these methods have shown competitive performance compared to classic causal discovery approaches, they are limited to working with only observational data. Our proposed method Structure Discovery from Interventions (SDI) is among the first causal discovery methods based on neural networks that utilize both observational and interventional data. Several subsequent works have built upon our proposed method, SDI. For instance, some works have explored the use of different gradient estimates (Brouillard et al., 2020; Lippe et al., 2021), while others have applied SDI in the active causal learning setting to determine where to intervene (Scherrer et al., 2021).

We introduce a novel neural network-based method for causal discovery called Structure Discovery from Interventions (SDI). Unlike previous methods, SDI utilizes both observational and interventional data. Our model outperforms previous methods on both synthetic and naturalistic Causal Bayesian Networks (CBNs). In addition, interventions in the real world may be performed by unknown agents, making them unknown interventions. Our model, SDI, can handle such unknown interventions by predicting the index of the node where the intervention may have occurred.

In some cases, the graph structure may be partially provided but needs to be completed. An example is protein structure learning in biology, where we may have definite knowledge about some parts of the protein-protein interaction structure but need to fill out other parts (Glymour et al., 2019). We refer to this as partial graph completion. In this setting, we evaluate our model and demonstrate its strong performance.

To summarize, the main contributions of the paper is as follows. 1) We propose a novel neural network based causal discovery algorithm SDI, which leverages both observational and interventional data. 2) We show that SDI outperforms previous methods on both synthetic data, as well as naturalistic data. 3) We show that SDI is effective even in the presence of unknown interventions. 4) We demonstrate that SDI generalizes well to unseen interventions. 5) We show that SDI is effective for partial graph discovery.

## 2 Background

**Causal modeling.** A Structural Causal Model (SCM) (Pearl, 1995; Peters et al., 2017) over a finite number $M$ of random variables $X_i$ is a set of structural assignments

$$X_i := f_i(X_{pa(i,C)}, N_i), \quad \forall i \in \{0, \ldots, M-1\} \tag{1}$$

where $N_i$ is jointly-independent noise and $pa(i, C)$ is the set of parents (direct causes) of variable $i$ under hypothesized configuration $C$ of the SCM directed acyclic graph, i.e., $C \in \{0, 1\}^{M \times M}$, with $c_{ij} = 1$ if node $i$ has node $j$ as a parent (equivalently, $X_j \in X_{pa(i,C)}$; i.e. $X_j$ is a direct cause of $X_i$). In our setup, all variables are observed.

**Intervention.** According to Eaton and Murphy (Eaton & Murphy, 2007b), there are several types of interventions that may be available, including:

*No intervention:* where only observational data is obtained from the ground truth model.

*Hard/perfect intervention:* where the value of one or more variables is fixed, and then ancestral sampling is performed on the remaining variables.

*Soft/imperfect intervention:* where the conditional distribution of the variable on which the intervention is performed is altered.

*Unknown intervention:* where the learner does not know which variable was directly affected by the intervention.

In this work, we permit interventions on all variables. Furthermore, we use soft interventions because they include hard interventions as a limiting case and are thus more general.

**Identifiability.** In a purely-observational setting, it is known that causal graphs can be distinguished only up to a Markov equivalence class. In order to identify the true causal graph structure, intervention data is needed (Eberhardt et al., 2006; Eberhardt & Scheines, 2007; Eberhardt et al., 2012; Eberhardt, 2012). When an infinite amount of single-node interventional data samples are available, the underlying causal graph can be identified (Eberhardt et al., 2006). Empirically, we have observed that even with a small number of samples per intervention, our model can make reasonable predictions, and its performance improves as more data becomes available.

**Structure discovery using continuous optimization.** Structure discovery is a search problem though the super-exponentially sized space of all possible directed acyclic graphs (DAGs). Previous continuous-optimization structure learning works (Zheng et al., 2018; Yu et al., 2019; Lachapelle et al., 2019) mitigate the problem of searching in the super-exponential set of graph structures by considering the degree to which a hypothesis graph violates "DAG-ness" as an additional penalty to be optimized. Despite the success of these methods, their ability for causal discovery is significantly limited because they can only operate on observational data.

## 3 Related Work

The recovery of the underlying structural causal graph from observational and interventional data is a fundamental problem (Pearl, 1995; 2009; Spirtes et al., 2000). Different approaches have been studied: score-based, constraint-based, asymmetry-based and continuous optimization methods. Score-based methods search through the space of all possible directed acyclic graphs (DAGs) representing the causal structure based on some form of scoring function for network structures (Heckerman et al., 1995; Chickering, 2002; Tsamardinos et al., 2006; Hauser & Bühlmann, 2012; Goudet et al., 2017; Cooper & Yoo, 1999; Zhu & Chen, 2019). Out of these approaches, only Hauser & Bühlmann (2012); Goudet et al. (2017); Cooper & Yoo (1999) can handle interventional data. Constraint-based methods (Spirtes et al., 2000; Sun et al., 2007; Zhang et al., 2012; Monti et al., 2019) infer the DAG by analyzing conditional independences in the data, none of these handles interventional data. Eaton & Murphy (2007c) use dynamic programming techniques to accelerate Markov Chain Monte Carlo (MCMC) sampling in a Bayesian approach to structure learning for discrete variable DAGs, the method utilizes interventional data. Asymmetry-based methods (Shimizu et al., 2006; Hoyer et al., 2009; Peters et al., 2011; Daniusis et al., 2012; Budhathoki & Vreeken, 2017; Mitrovic et al., 2018) assume asymmetry between cause and effect in the data and try to use this information to estimate the causal structure, none of these methods can handle interventional data. Peters et al. (2016); Ghassami et al. (2017); Rojas-Carulla et al. (2018) exploit invariance across environments to infer causal structure, which faces difficulty scaling due to the iteration over the super-exponential set of possible graphs, all three methods can handle interventional data. Mooij et al. (2016) propose a modelling framework that leverages existing methods while being more powerful and applicable to a wider range of settings, however, it can only handle observational data. Recently, (Zheng et al., 2018; Yu et al., 2019; Lachapelle et al., 2019) framed the structure search as a continuous optimization problem, however, the methods only uses observational data and are non-trivial to extend to interventional data. In our paper, we present a method that uses continuous optimization methods that works on both observational and interventional data.

For interventional data, it is often assumed that the models have access to full intervention information, which is rare in the real world. Rothenhäusler et al. (2015) have investigated the case of additive shift interventions, while Eaton & Murphy (2007b) have examined the situation where the targets of experimental interventions are imperfect or uncertain. This is different from our setting where the intervention is unknown to start with and is assumed to arise from other agents and the environment. Bengio et al. (2019) propose a meta-learning framework for learning causal models from interventional data. However, the method Bengio et al. (2019) explicitly models every possible set of parents for every child variable and attempts to distinguish the best amongst the combinatorially many such parent sets. It cannot scale beyond trivial graphs and only 2 variable experiments are presented in the paper. Several subsequent studies have built upon our proposed method SDI. Some of these studies have modified the gradient estimate (Brouillard et al., 2020; Lippe et al., 2021), while others have utilized SDI in the active causal learning setting to determine where to intervene (Scherrer et al., 2021).

Learning based methods have been proposed (Guyon, 2013; 2014; Lopez-Paz et al., 2015) which only handles observational data. There also exist recent approaches using the generalization ability of neural networks to learn causal signals from purely observational data (Kalainathan et al., 2018; Goudet et al., 2018). Neural network methods equipped with learned masks, such as (Ivanov et al., 2018; Li et al., 2019; Yoon et al., 2018; Douglas et al., 2017), exist in the literature, but only a few (Kalainathan et al., 2018) have been adapted to causal inference. This last work is, however, tailored for causal inference on continuous variables and from observations only. Adapting it to a discrete-variable setting is made difficult by its use of a Generative Adversarial Network (GAN) (Goodfellow et al., 2014) framework.

## 4 Structure Discovery from Interventions (SDI)

The proposed model, SDI, is a neural-network-based approach for causal discovery that integrates both observational and interventional data. The model models the causal mechanisms (conditional probability distributions) and the causal graph structure using two distinct sets of parameters. We refer to the parameters that capture the conditional probability between variables and their causal parents as *functional parameters*, denoted by $\theta$. The other set of parameters that describe the causal graph structure are referred to as *structural parameters*, represented by $\gamma$. The structural parameters $\gamma$ correspond to the learned adjacency matrix that encodes the graph structure of the SCM. Details given below.

**Parametrization.** Given a graph of $M$ variables, the structural parameter $\gamma$ is a matrix $\mathbb{R}^{M \times M}$ such that $\sigma(\gamma_{ij})$ is our belief in random variable $X_j$ being a direct cause of $X_i$, where $\sigma(x) = 1/(1 + \exp(-x))$ is the sigmoid function. The matrix $\sigma(\gamma)$ is thus a softened *adjacency matrix*.

The set of all functional parameters $\theta$ comprises the parameters $\theta_i$ that model the conditional probability distribution of $X_i$ given its parent set $X_{\text{pa}(i,C)}$, with $C \sim \text{Ber}(\sigma(\gamma))$ a hypothesized configuration of the SCM's DAG.

**Hypothesis.** According to the Independent Causal Mechanisms (ICM) principle (Schölkopf et al., 2012b), the causal generative process of a system's variables comprises autonomous modules that operate independently of one another. In probabilistic terms, this means that each variable's conditional distribution, given its causes (i.e., its mechanism), does not affect or influence the other mechanisms. Therefore, by having the SCM and knowledge of performed interventions, we can precisely predict unseen interventions. Based on this assumption, we hypothesize that the graph structure that better predicts unseen interventions is more likely to reflect the correct causal structure. This assumption has been verified experimentally for 2 variable cases in Bengio et al. (2019).

Following the aforementioned hypothesis, we adopt a two-phase training approach for our model, as illustrated in Figure 1. Since the structural and functional parameters are interdependent and influence each other, we train them in alternating phases, akin to a block coordinate descent optimization.

In the first phase, we train the functional parameters using observational data to infer the relationships between variables. In the second phase, we sample a few graphs from the model's current belief of the graph structure and evaluate their performance in predicting unseen intervention outcomes. We use this evaluation

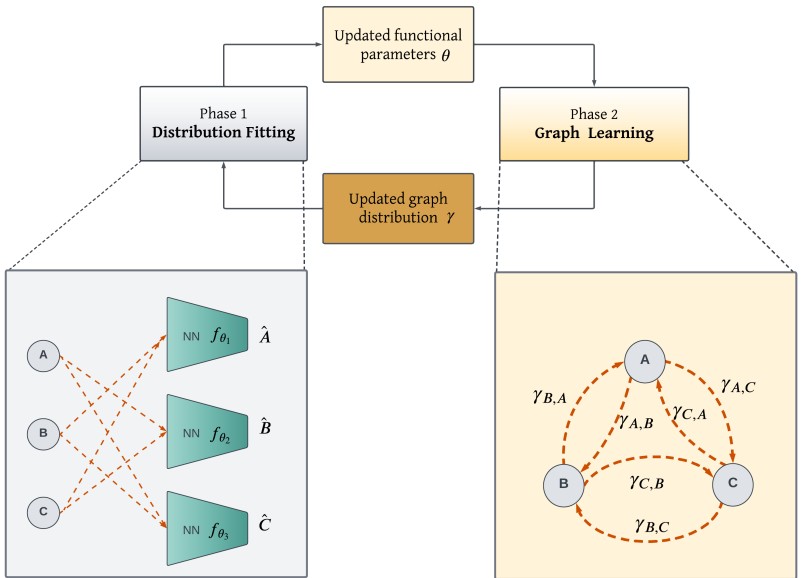

Figure 1: Our proposed method SDI. Phase 1 fits the MLP (functional) parameters on observational data. Phase 2 samples a small set of graphs under the model's current belief about the edge structure, and then scores these graphs against interventional data and assigns rewards according to graphs' ability to predict interventions. These rewards are then used to update the beliefs about the edge structure. The convergence of the method can be determined when the believed edge probabilities have reached saturation, approaching either 0 or 1. In our practical implementation, we train our model for $50,000$ steps and report the results.

as a reward signal to update our structure parameters $\gamma$. We will now elaborate on the aforementioned process in detail.

**Model.** Shown in Figure 2, our model consists of $M$ MLPs, where $M$ is the number of variables in the graph. Let $\theta_i = \{\texttt{W0}_i, \texttt{B0}_i, \texttt{W1}_i, \texttt{B1}_i\}$ define a stack of $M$ one-hidden-layer MLPs, one for each random variable $X_i$. A more appropriate model, such as a CNN, can be chosen using domain-specific knowledge; the primary advantage of using MLPs is that the hypothesized DAG configurations $c_{ij}$ can be readily used to mask the inputs of MLP $i$, as shown in Figure 2.

To force the structural equation $f_i$ corresponding to $X_i$ to rely exclusively on its direct ancestor set $\mathrm{pa}(i, C)$ under hypothesis adjacency matrix $C$ (See Eqn. 1), the one-hot input vector $X_j$ for variable $X_i$'s MLP is masked by the Boolean element $c_{ij}$. An example of the multi-MLP architecture with $M{=}3$ categorical variables of $N{=}2$ categories is shown in Figure 2. For more details, refer to Appendix A.4.

### 4.1 Phase 1: Distribution Fitting

During Phase 1, the functional parameters $\theta$ are trained to learn causal mechanisms (conditional probability distributions) by fitting them to observational data. To be specific, they maximize the likelihood of randomly drawn observational data under graphs randomly drawn from our current beliefs about the edge structure. We draw graph configurations $C_{ij} \sim \mathrm{Ber}(\sigma(\gamma_{ij}))$ and batches of observational data from the SCM, then maximize the log-likelihood of the batch under that configuration using stochastic gradient descent (SGD). The use of graph configurations sampling from Bernoulli distributions is analogous to dropout on the inputs of the functional models (in our implementation, MLPs), giving us an ensemble of neural networks that can model the observational data. To be specific, parameters $\theta$ is trained using maximum likelihood estimation on observation data $X$, such that $\theta^* = \mathrm{argmin}_\theta \mathcal{L}(\theta)$, where

$$L(\theta) = -\log P(X|C\,;\theta) \tag{2}$$

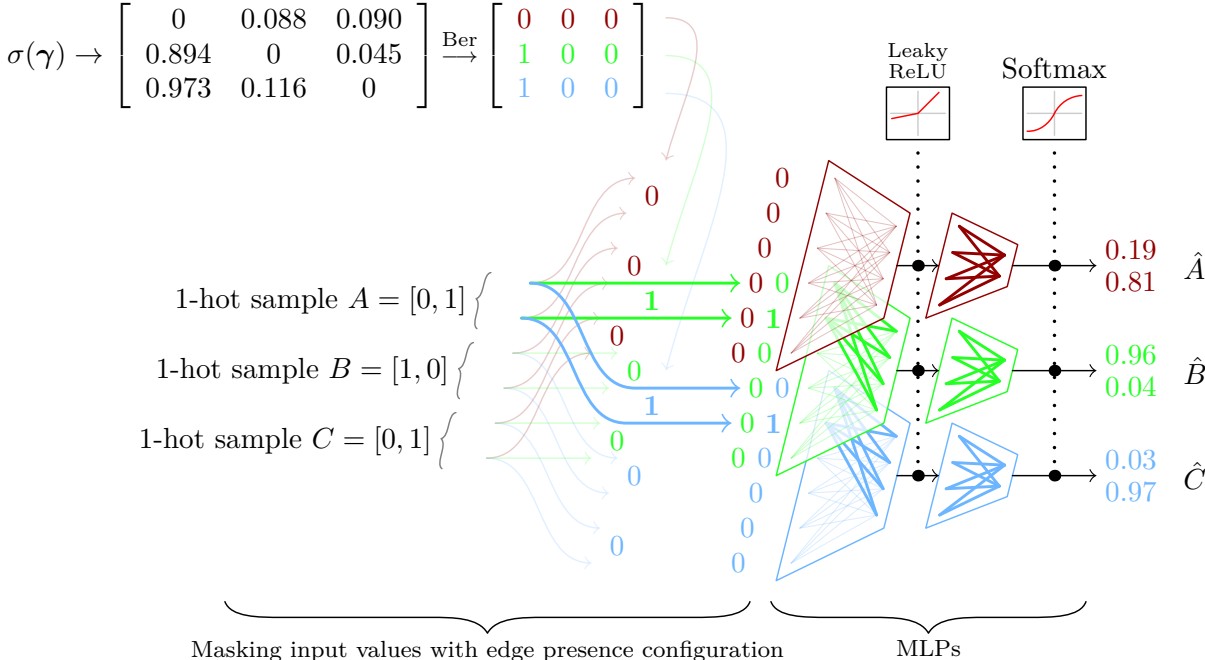

Figure 2: MLP Model Architecture for $M = 3$, $N = 2$ (fork3) SCM. The model computes the conditional probabilities of $A, B, C$ given their parents using a stack of three independent MLPs. The MLP input layer uses an adjacency matrix sampled from $\text{Ber}(\sigma(\gamma))$ as an input mask to force the model to make use only of parent nodes to predict their child node.

## 4.2 Phase 2: Graph Learning

During Phase 2, several graph configurations are sampled from the current learned edge beliefs parametrized by $\gamma$, and scored on data samples drawn from the intervened SCM.

To be more specific, we first sample $p$ graph configurations $C$ from the current and learned edge beliefs (structural parameters $\gamma$). We then obtain $k$ data samples $X$ from an intervention and evaluate the log-likelihood of the intervened data within the sampled graph configurations. This log-likelihood is denoted as $\log P(X|C;\theta)$. However, we make one modification: we mask the contribution of the intervened (or predicted-intervened) random variable $X_i$ to the total log-likelihood of the sample.

Since $X_i$ was intervened upon (using a Pearl do-operation, soft or hard), the values obtained for that variable should be treated as fixed rather than contributors to the total log-likelihood of the sample, and a correction gradient should not propagate through it (because the variable's CPT or MLP did not actually determine the outcome). It is assumed that the intervention is known at this stage, and we will explain in Section 4.2.1 how we predict the intervention when it is unknown.

### 4.2.1 Credit Assignment to Structural Parameters

The scores of the interventional data over various graph configurations are aggregated into a gradient for the structural parameters $\gamma$. Because a discrete Bernoulli random sampling process was used to sample graph configurations under which the log-likelihoods were computed, we require a gradient estimator to propagate gradient through to the $\gamma$ structural parameters. Several alternatives exist, but we adopt for this purpose the REINFORCE-like gradient estimator $g_{ij}$ with respect to $\gamma_{ij}$ proposed by Bengio et al. (2019):

$$g_{ij} = \frac{\sum_k (\sigma(\gamma_{ij}) - c_{ij}^{(k)}) \mathcal{L}_{C,i}^{(k)}(X)}{\sum_k \mathcal{L}_{C,i}^{(k)}(X)}, \quad \forall i,j \in \{0, \ldots, M-1\} \tag{3}$$

where the $^{(k)}$ superscript indicates the values obtained for the $k$-th draw of $C$ under the current edge beliefs parametrized by $\gamma$. The $\mathcal{L}_{C,i}^{(k)}(X)$ is the pseudo log-likelihood of variable $X_i$ in the data sample $X$ under the $k$'th configuration, $C^{(k)}$, drawn from our edge beliefs. Using the estimated gradient, we then update $\gamma$ with SGD, and return to Phase 1 of the continuous optimization process.

**Acyclic Constraint.** We include a regularization term $J_{\mathrm{DAG}}(\gamma)$ that penalizes length-2 cycles in the learned adjacency matrix $\sigma(\gamma)$, with a tunable strength $\lambda_{\mathrm{DAG}}$. The regularization term is $J_{\mathrm{DAG}}(\gamma) = \sum_{i \neq j} \cosh(\sigma(\gamma_{ij})\sigma(\gamma_{ji})), \quad \forall i,j \in \{0, \ldots, M-1\}$ and is derived from Zheng et al. (2018). The details of the derivation are in the Appendix. We explore several different values of $\lambda_{\mathrm{DAG}}$ and their effects in our experimental setup. Suppression of longer-length cycles was not found to be worthwhile for the increased computational expense.

**Predicting Intervention.** If the target of the intervention is known, then no prediction is needed. However, in the case of an unknown intervention, we employ a simple heuristic to *predict* it. During the initial stage of Phase 2 in iteration $t$, a small set of holdout interventional data samples is provided to the model. We then calculate the average log-likelihood for each individual variable $X_i$ across these samples. The variable $X_i$ exhibiting the most significant decline in log-likelihood is assumed to be the target of the intervention.

The rationale behind this approach is that if our model has effectively learned to represent the underlying causal graph, it should be capable of predicting the values of all variables $X_j$, except for the one that has undergone intervention. Initially, when our model has not yet acquired a reliable representation of the underlying causal graph, this prediction might be inaccurate. However, as training progresses and our model improves its predictive capabilities, the accuracy in predicting the intervention target becomes significantly enhanced. In section 6.4, we present the results of our causal discovery experiments based on either predicting the intervention for unknown interventions or having knowledge of the intervention target. We observe that the model's performance in predicting the intervention closely corresponds to having access to the ground truth.

## 5 Synthetic Data

This section discusses the generation process of synthetic data.

**Conditional Probability Distributions.** The conditional probability distributions (CPD) of the sythetic data are modeled by randomly initialized neural networks. All neural networks are 2 layered feed forward neural networks (MLPs) with Leaky ReLU activations between layers. The parameters of the neural network are initialized orthogonally within the range of $(-2.5, 2.5)$. This range was selected such that they output a non-trivial distribution. The biases are initialized uniformly between $(-1.1, 1.1)$. These values are selected to ensure that the CPDs are non-trivial and interesting, meaning they are neither almost deterministic nor uniformly distributed.

**Graph Structures.** In order to conduct a systematic analysis of our model's performance, we evaluate its performance on different graph structures. Our selection of synthetic graphs explores various extremes in the space of DAGs, stress-testing SDI. The `chain` graphs are the sparsest connected graphs possible, and are relatively easy to learn. The `bidiag` graphs are extensions of `chain` where there are 2-hops as well as single hops between nodes, doubling the number of edges and creating a meshed chain of forks and colliders. The `jungle` graphs are binary-tree-like graphs, but with each node connected directly to its grandparent in the tree as well. Half the nodes in a `jungle` graph are leaves, and the out-degree is up to 6. The `collider` graphs deliberately collide independent $M - 1$ ancestors into the last node; They stress maximum in-degree. Lastly, the `full` graphs are the maximally dense DAGs. All nodes are direct parents of all nodes below them

in the topological order. The maximum in- and out-degree are both $M - 1$. Please refer to Figure 3 for illustrative examples of 3-variable graphs, and consult Figure 6 for the complete collection of graphs.

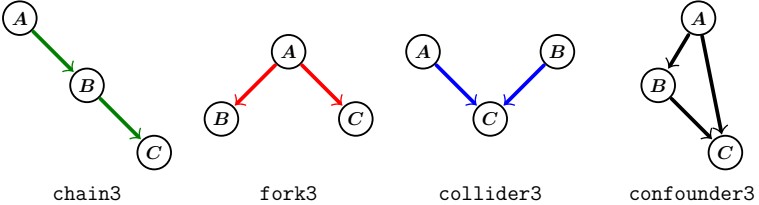

Figure 3: Example graphs for 3-variable connected DAG.

**Interventions.**  In all experiments, observational and interventional data are used. To generate interventional data, random and soft interventions are performed. The process begins by randomly selecting a variable, denoted as $X_i$, to be intervened on. Subsequently, a soft intervention is applied to the intervened variable $X_i$, replacing the conditional probability distribution $p(X_i|pa(X_i))$ with a different CPD denoted as $p'(X_i|pa(X_i))$. To be specific, the MLP (Multilayer Perceptron) parameters of the conditional distribution are reinitialized randomly within the range of (-2.5, 2.5), which aligns with the range of values used for the initial random initialization of the conditionals.

## 6  Experimental Setup and Results

We evaluate the performance of SDI across a range of experiments with increasing difficulties. We first evaluate SDI on a synthetic dataset where we have control over the number of variables and causal edges in the ground-truth SCM. This allows us to analyze the performance of the proposed method under various conditions. We then evaluate the proposed method on real-world datasets from the BnLearn dataset repository. We also consider the two variations: Recovering only part of the graph (when the rest is known), and exploiting knowledge of the intervention target.

The summary of our findings is: 1) We show strong results for graph recovery for all synthetic graphs in comparisons with other baselines, measured by Hamming distance. 2) The proposed method achieves high accuracy on partial graph recovery for large, real-world graphs. 3) The proposed method's intervention target prediction heuristic closes the gap between the known- and unknown-target intervention scenarios. 4) The proposed method generalizes well to unseen interventions. 5) The proposed method's time-to-solution scaling appears to be driven by the number of edges in the groundtruth graph moreso than the number of variables.

**Hyperparameters.**  Unless specified otherwise, we maintained identical hyperparameters for all experiments. In the subsequent paragraph, we examine the impact of DAG and sparsity penalties. The experiments involving SDI were executed for a total of 50,000 steps, as illustrated in Figure 10, and most experiments reached convergence within that timeframe. For additional information regarding the hyperparameter configuration, please refer to Appendix §A.5.

**Baseline Comparisons**  We compare SDI against a range of state-of-the-art methods, including both classic and neural network-based approaches. Specifically, we consider DAG-no-tears (Zheng et al., 2018), DAG-GNN (Yu et al., 2019), ICP (Peters et al., 2016), non-linear ICP (Heinze-Deml et al., 2018b), and (Eaton & Murphy, 2007b). Our comparison encompasses both synthetic and naturalistic data. DAG-no-tears and DAG-GNN are neural network-based causal discovery methods and can only handle observational data. On the other hand, ICP, non-linear ICP, and Eaton & Murphy (2007b) are designed for interventional data. Notably, Eaton & Murphy (2007b) addresses uncertain interventions, while ICP and non-linear ICP handle unknown interventions. However, none of these methods attempt to predict the intervention. All methods are provided with both observational and interventional data, which is identical to the data given to SDI, except for the methods that solely handle observational data, in which case only observational data is provided.

## 6.1 Results on Synthetic Data

We first evaluate the model's performance on synthetic data generated from multiple SCMs (Structural Causal Models) with specific and representative graph structures. These SCMs are randomly initialized using the procedure described in Section 5. Since the number of possible DAGs grows super-exponentially with the number of variables, for $M=4$ up to 13 a selection of representative and edge-case DAGs are chosen. `chainM` and `fullM` ($M=3$-13) are the minimally- and maximally-connected $M$-variable DAGs, while `treeM` and `jungleM` are tree-like intermediate graphs. `colliderM` is the $(M-1) \to 1$ collider graph. The details of the setup is in Appendix A.6.

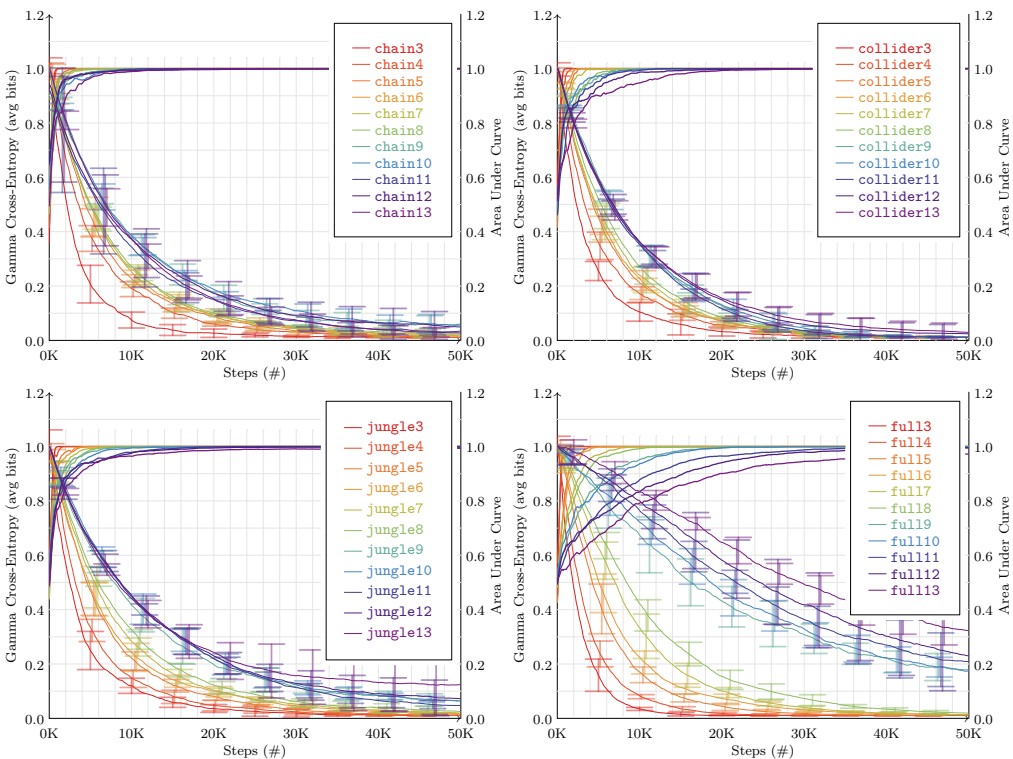

Figure 4: Cross entropy (CE) and Area-Under-Curve (AUC/AUROC) for edge probabilities of learned graph against ground-truth for synthetic SCMs. Error bars represent $\pm 1\sigma$ over PRNG seeds 1-5. **Left to right, up to down**: `chainM,jungleM,fullM`,$M = 3 \dots 8$ ($9 \dots 13$ in Appendix A.6.1). Graphs (3-13 variables) all learn perfectly with AUROC reaching 1.0. However, denser graphs (fullM) take longer to converge.

**Results.** The model can recover most synthetic DAGs with high accuracy, as measured by Structural Hamming Distance (SHD) between learned and ground-truth DAGs. Table 1 shows the hamming distance between the groundtruth and and the learned graphs for all methods, we threshold our edge beliefs at $\sigma(\gamma) = 0.5$ to derive a graph. As shown in the table, our proposed method SDI outperforms all other methods, and learns all graphs perfectly for 3 to 13 variables (excepting `full`).

In Figure 10, we present visualizations of the cross-entropy loss and AUROCs (Area Under the Receiver Operating Characteristic Curve) obtained during the training process of SDI. The continuous decrease in the cross-entropy loss indicates SDI's convergence towards the correct causal model. Notably, for graphs with up to 10 variables, the AUROCs consistently reach a value of 1.0, signifying perfect classification into edge and non-edge categories.

Note that Eaton & Murphy (2007b) runs out of memory for graphs larger than $M = 10$ because modelling of uncertain interventions is done using "shadow" random variables (as suggested by the authors), and thus

| Method | Asia | Sachs | collider | chain | jungle | collider | full |
|---|---|---|---|---|---|---|---|
| $M$ | 8 | 11 | 8 | 13 | 13 | 13 | 13 |
| **Zheng et al. (2018)** | 14 | 22 | 18 | 39 | 22 | 24 | 71 |
| **Yu et al. (2019)** | 10 | 19 | 7 | 14 | 16 | 12 | 77 |
| **Heinze-Deml et al. (2018b)** | 8 | 17 | 7 | 12 | 12 | 7 | 28 |
| **Peters et al. (2016)** | 5 | 17 | 2 | 2 | 8 | 2 | 16 |
| **Eaton & Murphy (2007a)** | 0 | OOM | 7 | OOM | OOM | OOM | OOM |
| **Proposed Method (sdi)** | 0 | 6 | 0 | 0 | 0 | 0 | 7 |

Table 1: **Baseline comparisons:** Structural Hamming Distance (SHD) (lower is better) for learned and ground-truth edges on various graphs from both synthetic and real datasets, compared to (Peters et al., 2016), (Heinze-Deml et al., 2018b), (Eaton & Murphy, 2007b), (Yu et al., 2019) and (Zheng et al., 2018). The proposed method (SDI) is run on random seeds $1-5$ and we pick the worst performing model out of the random seeds in the table. OOM: out of memory. Our proposed method correctly recovers the true causal graph, with the exception of Sachs and full 13, and it significantly outperforms all other baseline methods.

recovering the DAG internally requires solving a $d = 2M$-variable problem. Their method's extremely poor time- and space-scaling of $O(d2^d)$ makes it unusable beyond $d > 20$.

## 6.2 Results on Real-World Datasets: BnLearn

The Bayesian Network Repository (`www.bnlearn.com/bnrepository`) is a collection of commonly-used causal Bayesian networks from the literature, suitable for Bayesian and causal learning benchmarks. We evaluate the proposed method on the Earthquake (Korb & Nicholson, 2010), Cancer (Korb & Nicholson, 2010), Asia (Lauritzen & Spiegelhalter, 1988) and Sachs (Sachs et al., 2005) datasets ($M =$5, 5, 8 and 11-variables respectively, maximum in-degree 3) in the BnLearn dataset repository.

**Results.** As shown in Table 1, SDI perfectly recovers the DAG of Asia, while making a small number of errors (SHD=6) for Sachs (11-variables). It thus significantly outperforms all other baselines models. Figures 8 & 9 visualize what the model has learned at several stages of learning. Results for Cancer and Asia can be found in the appendices, Figure 15 and 16.

## 6.3 Generalization to Previously Unseen Interventions

It is often argued that machine learning approaches based purely on capturing joint distributions do not necessarily yield models that generalize to unseen experiments, since they do not explicitly model changes through interventions. By way of contrast, causal models use the concept of interventions to explicitly model changing environments and thus hold the promise of robustness even under distributional shifts (Pearl, 2009; Schölkopf et al., 2012a; Pe-

Table 2: **Evaluating the consequences of a previously unseen intervention:** (test log-likelihood under intervention)

| | fork3 | chain3 | confounder3 | collider3 |
|---|---|---|---|---|
| **Baseline** | -0.5036 | -0.4562 | -0.3628 | -0.5082 |
| **sdi** | -0.4502 | -0.3801 | -0.2819 | -0.4677 |

ters et al., 2017). To test the robustness of causal modelling to previously unseen interventions (new values for an intervened variable), we evaluate a well-trained causal model against a variant, non-causal model trained with $c_{ij} = 1$, $i \neq j$. An intervention is performed on the SCM, fresh interventional data is drawn from it, and the models, with knowledge of the intervention target, are asked to predict the other variables given their parents. The average log-likelihoods of the data under both models are computed and contrasted. The intervention variable's contribution to the log-likelihood is masked. For all 3-variable graphs (`chain3`, `fork3`, `collider3`, `confounder3`), the causal model attributes higher log-likelihood to the intervention distribution's

samples than the non-causal variant, thereby demonstrating causal models' superior generalization ability in transfer tasks. Table 2 collects these results.

## 6.4 Variant: Predicting interventions

In Phase 2 (described in §4.2), the contribution of the intervened (or predicted-intervened) random variable $X_i$ is masked from the total log-likelihood of the sample. To predict the intervention target variable, we employ a simple heuristic. Experimental results demonstrate that this heuristic performs well in

Table 3: **Intervention Prediction Accuracy:** (identify on which variable the intervention took place)

| 3 variables | 4 variables | 5 variables | 8 variables |
|---|---|---|---|
| 95 % | 93 % | 85 % | 71 % |

practice, yielding accurate predictions more frequently than chance alone (see Table 3). In this section, we conduct ablation studies to thoroughly examine the significance of intervention prediction. We compare several scenarios to evaluate their impact on the overall results.

In the first scenario, referred to as 'guessing intervention,' we randomly guess the intervention variable and subsequently mask out its contribution. The second scenario, termed 'no information on intervention,' involves not guessing the intervention variable at all, which means all variables, including the intervened-on variable, contribute to the log-likelihood used in Phase 2. The third scenario, 'predicting unknown intervention,' entails asking the model to predict the intervened variable and subsequently masking out its contribution. Lastly, in the 'known intervention' scenario, we provide the model with the ground truth information about the intervened variable, enabling the masking of its contribution.

Notably, both the first and second scenarios, where either the model randomly guesses the intervention or doesn't use any intervention information at all, result in a significant decline in model performance. This decline holds true even for graphs with only three variables, as demonstrated in Figure 5 (Left). Conversely, when training SDI with intervention prediction, the model's performance closely aligns with training that incorporates the ground-truth intervention. This alignment persists even for larger graphs comprising seven variables, as depicted in Figure 5 (Right)

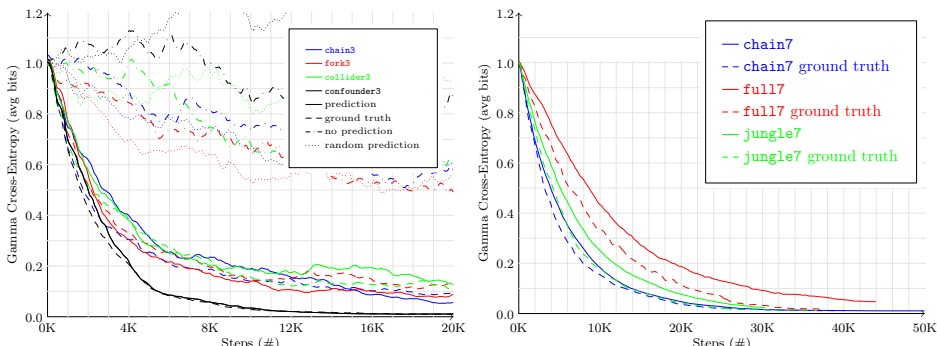

Figure 5: **Ablation Study of Intervention Prediction** Cross-entropy loss over time on multiple graphs and intervention prediction modes. **Left**: All 3-variable graphs. Solid/dashed lines: Ground-truth & Prediction strategies. Dotted lines: Random- & No-Prediction strategies. Training with prediction closely tracks ground-truth. **Right**: Comparison for 7-variable graphs, ground-truth against prediction strategy. Training with predicted interventions still closely tracks using ground-truth interventional information at larger scales.

## 6.5 Variant: Partial Graph Recovery

Instead of learning causal structures *de novo*, we may have partial information about the black-box SCM and may only need to fill in missing information. An example is protein structure discovery in biology (Glymour

et al., 2019), where some causal relationships have been definitely established and others remain open hypotheses. This is an easier task compared to full graph recovery, since the model only has to search for missing edges.

We evaluate SDI on Barley (Kristensen & Rasmussen, 2002) ($M = 48$) and Alarm (Beinlich et al., 1989) ($M = 37$) from the BnLearn repository. The model is asked to predict 50 edges from Barley and 40 edges from Alarm. The model reached $\geq 90\%$ accuracy on both datasets, as shown in Table 4.

Among the methods evaluated in Table 1, the top three performers were ICP, non-linear ICP, and the method proposed by Eaton & Murphy (2007a). However, it should be noted that these methods are not scalable to larger graphs, specifically in the case of Barley and Alarm datasets. Consequently, a direct comparison with these methods was not feasible.

Table 4: **Partial Graph Recovery** on Alarm (Beinlich et al., 1989) and Barley (Kristensen & Rasmussen, 2002). The model is asked to predict 50 edges in Barley and 40 edges in Alarm. The accuracy is measured in Structural Hamming Distance (SHD). SDI achieved over 90% accuracy on both graphs.

| Graph | **Alarm** | **Barley** |
|---|---|---|
| Number of variables | 37 | 48 |
| Total Edges | 46 | 84 |
| Edges to recover | 40 | 50 |
| Recovered Edges | 37 | 45 |
| Errors (in SHD) | 3 | 5 |

### 6.6 Ablation and analysis

As shown in Figure 11, larger graphs (such as $M > 6$) and denser graphs (such as `full8`) are progressively more difficult to learn. For denser graphs, the learned models have higher sample complexity, higher variance and slightly worse results. Refer to Appendix §A.9 for complete results on all graphs.

**Importance of regularization.** Valid configurations $C$ for a causal model are expected to be a) sparse and b) acyclic. To promote such solutions, we introduce DAG and sparsity regularization with tunable hyperparameters. For all experiments, we set the DAG penalty to 0.5 and sparsity penalty to 0.1. We run ablation studies on different values of the regularizers and study their effect. We find that smaller graphs are less sensitive to different values of regularizer than larger graphs. For details, refer to Appendix §A.13.

**Importance of dropout.** To train functional parameter for an observational distribution, sampling adjacency matrices is required. We "drop out" each edge (with a probability of $\sigma(\gamma)$) in our experiments during functional parameter training of the conditional distributions of the SCM. Please refer to Appendix §A.14 for a more detailed analysis.

## 7 Conclusion

In this work, we introduced an experimentally successful method (SDI) for causal structure discovery using continuous optimization, combining information from both observational and interventional data. We show in experiments that it can recover true causal structure, that it generalizes well to unseen interventions, that it compares very well against the start-of-the-art causal discovery methods on real world datasets, and that it scales even better on problems where only part of the graph is known.

## Acknowledgements

This research was conducted while the first author was a graduate student at Mila. The authors would like to acknowledge the support of the following agencies for research funding and computing support: NSERC, Compute Canada, the Canada Research Chairs, CIFAR, and Samsung. We would like to thank Lars Buesing, Nasim Rahaman and Rémi Le Priol for useful feedback and discussions.

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
