# OpenReview forum: "Neural Causal Structure Discovery from Interventions"
_TMLR — Accepted by TMLR_

### Review · Reviewer_DbbW · 2023-04-29

**Summary Of Contributions:**

A method for identifying the structural causal graph is proposed. The method utilizes both observational and interventional data. It iterates between two stages. In one stage, the functional parameters (i.e., the parameters of neural nets that fit the conditional distributions) are trained using observational data. In the other stage, the structural parameters (i.e., the elements of the adjacency matrix) are trained using interventional data. The nice performance of the proposed method is empirically confirmed not only with synthetic SCMs but also with more realistic SCMs.

**Audience:**

Yes

**Claims And Evidence:**

Yes

**Requested Changes:**

1. Please address **W1**. Although there is a pseudocode in the appendix, I would recommend putting the core mathematical descriptions of the training process in the main text (Sections 4.1 and 4.2), too.

2. Please address **W2**. It would be nice if the authors could elaborate more on why the proposed heuristic is expected to work.

3. Please address **W3**. For example, what kind of information do the baseline methods in Table 1 use? Do they use both observational and interventional data as the proposed method does? (Maybe I missed some descriptions already there in other parts of the paper, though.)

Below are minor points.

4. Throughout the paper, please re-check the use of latex commands \citep and \citet (or something equivalent). Sometimes \citet is used when \citep should be.

5. Please also re-check the latex cross reference. "??" appears sometimes, especially in the supplementary material.

6. In Page 5, "... our model consists of $N$ MLPs, where $N$ is the number of variables in the graph": Do you mean $M$ instead of $N$?

7. Are the estimated causal mechanism (i.e., functional parameters) somehow referred to in the experiments? In my understanding, the performance evaluation is on the structural parameters only (please correct me if I'm wrong, which is likely). I wondered how the estimated causal mechanism could be useful.

**Strengths And Weaknesses:**

### Strengths

**S1**. The paper is well written. The motivation is clearly stated. The background and related work sections are useful.

**S2**. The strong performance of the proposed method is shown for different datasets.

**S3**. Not only the main structure identification problem but also other problems (the unknown intervention problem and the partial recovery problem) are addressed.

### Weaknesses

**W1**. The training process in Sections 4.1 and 4.2 are not mathematically clear. I could understand the rough idea but am not sure what happens there exactly.

**W2**. The description of the method for "Predicting Intervention" in Section 4.2.1 is not clear.

**W3**. The explanation of the baseline methods used in the experiment is not sufficient.

---

> ### Author Response · Authors · 2023-06-14
> **Response to Reviewer DbbW**
>
> We would like to thank the reviewer for their thorough and valuable feedback which will help us in improving our paper. We are grateful for the interest in our proposed method and appreciate their support for the writing and the results. We have carefully considered their comments and have made the following changes to our paper:
>
> **Regarding W1:  The training process in Sections 4.1 and 4.2 are not mathematically clear. I could understand the rough idea but am not sure what happens there exactly.**
>
> We are grateful to the reviewer for highlighting their concerns. We have revised 4.1 and 4.2 to enhance the mathematical clarity of these sections.
>
> Specifically, in Section 4.1, we have now included the explicit equation for the loss function, which helps to  provide a precise mathematical representation of the training process. In Section 4.2, we have incorporated mathematical formulations for various terminologies, such as the loglikelihood of interventional distribution. By including these formulations, we hope to provide a more rigorous and detailed explanation of the underlying mathematics and calculations for our method.
> We hope that these revisions effectively address the reviewer's concerns and provide the necessary clarity. We are open to incorporating any additional mathematical details if the reviewer identifies any further areas where clarity is needed.
>
> **Regarding W2: The description of the method for "Predicting Intervention" in Section 4.2.1 is not clear.**
>
> We appreciate the reviewer for pointing this out. We have updated section 4.2.1 ("predicting intervention"). The revised section now provides a more detailed and accurate description of the method employed for predicting interventions. We hope these modifications adequately address the concern raised by the reviewer.
>
> **Regarding W3: The explanation of the baseline methods used in the experiment is not sufficient.**
>
> We thank the reviewer for bringing this to our attention. we have included a subsection titled "Baseline Comparisons" in Section 6 of the paper. This subsection provides a explanation of our process for selecting competing methods and outlines how these methods are utilized.
>
> **Regarding typos and missing references. **
>
> We thank the reviewer for pointing this our. We have rectified the identified typos and missing references pointed out by the reviewers.
>
> **Regarding “Are the estimated causal mechanism (i.e., functional parameters) somehow referred to in the experiments?" **
>
> We appreciate the reviewer’s question. We did not focus on evaluating of causal mechanisms within the training data, which comprises intervention and observation data that the model has encountered. However, we conducted analyses in Section 6.3 to specifically assess the model's ability to generalize to previously unseen interventions. These experiments are designed to incorporate and evaluate the utilization of causal structure, as well as causal mechanisms.
>
> We sincerely appreciate the reviewer's valuable input, which has allowed us to improve the quality of our paper.

---

> > ### Comment · Reviewer_DbbW · 2023-06-20
> >
> > Thank you for the clarification. I'm basically now good with the revised version. Meanwhile as a general advice, it would have been much more helpful if the authors had highlighted the revised/added parts of the manuscripts in different colors.

---

### Review · Reviewer_h2Vy · 2023-05-14

**Summary Of Contributions:**

The paper proposes a method to estimate structural causal models (both the causal DAG and the functional relationship among variables) from observational and interventional data. The method is an alternating optimization, with two sets of decision variables: the first set parametrizes the causal DAG, and the second set parametrizes the functional relationships. Both phases of the alternating optimization is based on maximum likelihood. Experiments on small structural causal models demonstrate give evidence that that the method can successfully estimate structural causal models.

**Audience:**

Yes

**Claims And Evidence:**

Yes

**Requested Changes:**

Primary substantive changes:
* Explain the reasoning behind choosing the methods in Table 1 as competitors methods.
  * The paper's Related Work section seemingly lists at least a dozen papers on estimating SCM, but Table 1 compares the proposed method with only 5 such methods. It is possible that the chosen competitor methods are the best-performing out of their lines of work: if so, this should be explained in clear detail.

* Revise the first paragraph of Related Work section to clarify which methods can handle both observational and interventional data
  * Currently, the first two sentences in this paragraph suggests that the papers mentioned in the paragraph can handle both types of data. However, the last sentence "Recently, (Zheng et al., 2018; Yu et al., 2019; Lachapelle et al., 2019) framed the structure search as a continuous ...[] ... interventional data.' contradicts this assessment.

* Clarify the assumptions that enable causal graph learning from just one intervention
  * Right before Section 5.1, the paper says "In all experiments, at most one (soft) intervention is concurrently performed." For general SCMs with more than one parent-child edge, it seems unlikely that just one intervention is sufficient in recovering the underlying causal graph. I am willing to be convinced that this intuition is incorrect, however.

Secondary substantive changes:
* Explain the omission of the competitor methods in Table 1 from Table 4.
  * The Related Work section does not mention "inability to apply to partial graph recovery" as a limitation of existing works. If this were the case, it would be great to have that explicitly spelled out.

* Compare the runtimes of various methods
  * Based on Appendix A.11, the proposed method is seemingly highly time consuming, since many observations are needed to accurately estimate the causal graph. What do the runtime of the competitor methods look like? I do not think including runtime information will be detrimental to the proposed method.

Clarify writing and figures:
* Clarify if there has been an existing preprint on the SDI paper: if so, what is the relationship between the current paper and the existing preprint?
  * The sentence "Several subsequent studies have built upon our proposed method SDI" suggests that there is an existing paper on SDI. It is unclear if the current manuscript is a) the same paper or b) an extension. If it were an extension, I am not clear what parts are known versus what parts are new.
* After Equation (2), define the quantity $L_{C,i}^{(k)}(X)$. This notation is currently used without a definition.
* In Figure 3, increase the font sizes and remove the visual clutter. There are perhaps too many curves being plotted - although the colors help distinguish the curves, it is still difficulty to tell apart the curves.

Changes relating to references and citations:
* Fill in the missing reference in Section 5.6.
* Add references to support the sentence "An example is protein structure learning in biology ...". This sentence appears in the Introduction and Section 5.6.
* Add citation for the bnlearn package / dataset repository
* Replace the duplicate reference


**Strengths And Weaknesses:**

Strengths:
- Diagrams (Figure 1 and Figure 2) explain the estimation procedure visually.
- Experiments are conducted on different families of structural causal models (SCM). They showing that the estimation procedure can handle a diverse set of situations.
- The appendix contains sensitivity studies of the estimation accuracy with respect to the estimation procedure's hyperparameters.

Weaknesses:
- The paper does not explain the choice of competing methods included in the experiments. It maybe missing important competitors in the literature review and experimental results. Hence, the evidence to support the claim "our method achieves strong benchmark results on various structure learning tasks" (mentioned in the abstract, among other places) is not strong.
- There are places in which the quality of writing should be improved (please see Requested Changes).

---

> ### Author Response · Authors · 2023-06-13
> **Response to Reviewer h2Vy**
>
> We very much appreciate the reviewer’s time and feedback. We have updated our paper to address the reviewer’s concern.
>
> **Regarding “The paper does not explain the choice of competing methods included in the experiments…”**
>
> We appreciate the reviewer for pointing this out. We have included a subsection in the paper in Section 6, titled "Baseline Comparisons." This subsection explains how we chose the competing methods. In short, we compared a range of state-of-the-art methods for causal discovery, ranging from classic to neural network-based causal discovery methods.
>
> **Regarding “Clarify the assumptions that enable causal graph learning from just one intervention.”**
>
> We appreciate the reviewer for bringing this to our attention. We apologize for any confusion that may have been caused by our original wording. We meant to say that each interventional sample contains only one intervention.
> To clarify, our method learns to induce causal graphs from a dataset that contains multiple interventional samples, as long as each interventional sample contains only one intervention. We have updated our paper to reflect this clarification.
> In Section 5, we have added a new subsection titled Interventions. In this subsection, we discuss how interventions are performed and the assumptions that are necessary for our method to learn causal graphs from a dataset that contains multiple interventional samples.
>
> **Regarding “Revise the first paragraph of the Related Work section to clarify which methods can handle both observational and interventional data”.**
>
> We thank the reviewer for bringing this to our attention. We have updated the related works section to clarify which methods can handle interventional data.
>
> **Explain the omission of the competitor methods in Table 1 from Table 4.**
>
> e thank the reviewer for bringing this to our attention. We have included a discussion in Section 6.5 to explain why the competitor methods in Table 1 were omitted from Table 4. Specifically, we found that the top three performers in Table 1, ICP, nonlinear ICP, and the method proposed by Eaton and Murphy (2007a), were not scalable to larger graphs. As a result, we were unable to compare our method to these methods on the Barley and Alarm datasets, which are both larger graphs.
>
> **Regarding “Compare the runtimes of various methods”.**
>
> We thank the reviewer for bringing this to our attention. We have included a section on runtimes for all methods on 10-node graphs in Appendix Section A.12. The runtimes are as follows: Our proposed method \gls{SDI}, takes between 4 to 5 hours for training. In comparison, DAG-GNN \citep{yu2019dag} requires between 2 to 3 hours, and DAG-Notears takes around half an hour. The training time for ICP \citep{peters2016causal} is approximately 4 hours, and the non-linear ICP \citep{heinze2018causal} also takes about 4 hours. Additionally, \citet{eaton2007belief} requires approximately 12 hours for training.
>
> **Regarding “Clarify if there has been an existing preprint on the SDI paper: if so, what is the relationship between the current paper and the existing preprint?”**
>
> This is the original paper on SDI.
>
> **Regarding “Figure 3, increase the font sizes and remove the visual clutter. ”**
>
> We appreciate the reviewer for pointing this out. To enhance readability, we have increased the font sizes within the figure. Additionally, we have made the figure larger in order to provide a clearer representation of the data. Furthermore, in the final revision of the paper, we will visualize the CE loss and the AUROC in separate figures. This separation will contribute to a more organized and comprehensible presentation of the results.
>
> **Regarding “Defining the L_{c,i}^{k}”.**
>
> We thank the reviewer for pointing this out. This is defined in paragraph after equation (3) in section 4.2.1.
>
> **Regarding “References and citations”.**
>
> We have updated our paper to include the references and citations.
>
>
> We sincerely appreciate the reviewer's valuable input, which has allowed us to improve the quality of our paper.

---

> > ### Comment · Reviewer_h2Vy · 2023-07-04
> > **Rebuttal Acknowledgment**
> >
> > I thank the reviewers for their response. I am happy with the current state of the paper.

---

### Review · Reviewer_ZoQW · 2023-05-17

**Summary Of Contributions:**

This manuscript uses a sampling procedures and neural networks to more efficiently learn a structured causal model.  The primary contributions are methodological, and the approach appears to empirically outperform competing approaches.  The main focus is on using interventional data, so the algorithm is denoted as Structure Discovery from Interventions (SDI).  SDI follows a reasonable multi-step loop to effectively learn structures.

**Audience:**

Yes

**Broader Impact Concerns:**

Not necessary.

**Claims And Evidence:**

No

**Requested Changes:**

Critical:
The experimental section needs significantly more detail on how the data generation procedure worked, how the interventions were generated both for the proposed model and for competing models, and greater details on how competing models were used.  As it is now, I could not reproduce the results because I do not have enough details.  Without this, I cannot mark the claims and evidence below as yes.

Above, in minor weaknesses, I also highlighted several challenges in the interpretation of the provided results, namely Table 2 and Table 4.  Please provide context on the importance/significance of these results.

Make clarifications/fix typos as noted in minor weaknesses.

Not critical but would improve the work:
Moving many of the useful results from the appendix to the main draft would improve the flow of the manuscript.


**Strengths And Weaknesses:**

Strengths:
This algorithm appears to work well empirically, and the intuition behind the algorithm is straightforward.  The method's individual steps all make sense, and they appear to combine well together to make an effective algorithm for structured causal models.

Weaknesses:
First, there are weaknesses in the lack of theory supporting the proposed algorithm, and the computational complexity is still quite high, even if it is better than some competing approaches.  I do not view these challenges as easily addressable or necessary to solve in the context of this work, and instead focus below on addressable concerns.

Major Addressable Weaknesses:
The data generation procedure is not clear enough in the results, as many details are lacking.  In particular, it is unclear how the interventions work.  The detail in the appendix minimally states, “We perform our intervention by first randomly selecting which variable to intervene on, then soft-intervening on it.” Without this information, it is not clear how to evaluate the many claims in the experiments.  Additionally, how many interventions are available? It sounds like the authors are allowing new interventions every time during phase II of their training (step 11 of Algorithm 1), which is an unrealistic scenario.  How does this compare to the information provided to the competing models? Without significantly more information on the data generation and details on how the experiments and comparisons were run, I could not reproduce or completely validate the results.

Minor Addressable Weaknesses:
The results section appears to be written trying to save space by putting many necessary results in the appendix.  I would suggest putting all key results in the main paper, rather than repeatedly referencing the appendix.  This would improve flow.

M & N are presented in a confusing manner: “Given a graph of M variables” & “N is the number of variables in the graph”. Please correct.

“If the believed edge probabilities have all saturated near 0 or 1, the method has converged.” -> tricky convergence condition.  Please elaborate.  It sounds like from the experiments the algorithm is just run as long as possible.

In 4.2, you should explicitly state that the sampled graphs are learned.

The scale and importance of Table 2 is unclear.  How much better is this method?

Several references are the wrong format (e.g., end of paragraph 2).

ICM is not defined.

Figure 3 is messy and difficult to read since the line formats are the same for both metrics.  I would suggest splitting into two figures.

Section 5.5 is unclear, as it is not well enough described what the known and unknown interventional generation entails.

Section 5.6 lacks comparisons, making it difficult to evaluate the quality of the method.

Under “predicting intervention,” there is this phrase: “A small number of interventional data samples are drawn from the SCM and more graphs are sampled from our current edge beliefs.”  I don’t know what the authors meant by that.

I cannot find what interventions were used in Section 5.1 in the main paper or appendix.  As this is necessary for evaluation, it needs to be included in the main manuscript.

Some references are appearing as ?? (e.g., right below A.1)

Figure 5’s images are pixelized.

Small grammar things (e.g. “The” in A.10 should not be capitalized)

---

> ### Author Response · Authors · 2023-06-13
> **Response to Reviewer ZoQW**
>
> We would like to thank the reviewer for their thorough and valuable feedback which will help us considerably in improving our paper. We have carefully considered their comments and have made the following changes to our paper:
>
> **Regarding  “The data generation procedure is not clear enough in the result..  Without significantly more information on the data generation and details on how the experiments and comparisons were run, I could not reproduce or completely validate the results.”**
>
> We thank the reviewer for their feedback. We have updated our paper to include a more detailed description of the data generation process, including the specific interventions used. This information can be found in Section 5 (Synthetic Data). We have also added a discussion of how the comparisons between different methods were conducted, including the data that was given to each baseline method. This information can be found under the paragraph for “Baseline Comparisons” in Section 6.
>
> **Regarding “ The results section appears to be written trying to save space by putting many necessary results in the appendix. ”**
>
> We thank the reviewer for their valuable feedback. We have incorporated their suggestions and made the following changes to the paper:
> - We have moved the data generation process described in Section 5 back into the main paper.
> - We have moved Figure 5 back into the main paper.
> - We have moved the main results on the synthetic dataset (what was Figure 10 in the appendix) and merged it with Figure 3 in the first submission. We have made the figures larger so that they are easier to see and understand.
>
> We hope these changes will improve the clarity and readability of the paper. We appreciate the reviewer's help in making our paper better.
>
> **Regarding the typos, grammar fixes, and missing references mentioned by the reviewer, we have fixed all of them. We have also added a list of all the references cited in the paper in the bibliography. Specifically, we addressed the following comments:**
>
> - "M & N are presented in a confusing manner:" We have fixed this.
> - "In 4.2, you should explicitly state that the sampled graphs are learned." I added a sentence to Section 4.2 that states that the sampled graphs are learned.
> - "ICM is not defined." I added a definition of ICM to Section 3.1.
> - "Some references are appearing as ?? (e.g., right below A.1)" I fixed the references that were appearing as "??."
> - "Small grammar things (e.g. “The” in A.10 should not be capitalized)" I corrected the small grammar errors, such as the capitalization of "The" in Section A.10.
>
> **Regarding “Section 5.5 is unclear, as it is not well enough described what the known and unknown interventional generation entails.**
>
> We thank the reviewer for pointing out that Section 5.5 was unclear. We have updated what is now Section 6.4 to provide a better description of known and unknown interventional generation. Specifically, we added the following sentences to clarify the difference between known and unknown interventions:
> - Guessing intervention: We randomly predict the intervention variable and subsequently mask out its contribution.
> - No information on intervention: We do not guess the intervention variable at all, which means all variables, including the intervened-on variable, contribute to the log-likelihood used in Phase 2.
> - Predicting unknown intervention: We ask the model to predict the intervened variable and subsequently mask out its contribution.
> - Known intervention: We provide the model with the ground truth information about the intervened variable, enabling the masking of its contribution.
>
> **We appreciate the reviewer's feedback on Figure 5. We have remade Figure 6 (previously Figure 5) to improve the clarity of the images. The new figures are now in higher resolution and should be easier to read and understand.**

---

### Decision · Action_Editors · 2023-07-13

**Recommendation:** Accept as is

**Comment:**

The paper proposes a new techniques for causal discovery from data that also contains interventions.  The method is evaluated on problems of moderate scale, and shown to outperform both classical and neural network based baselines. Reviewers raised some issues around the choice of baselines, around the specific data generating process being assumed, around the assumptions and nature of the interventions and how these permitted, theoretically, the identification of causal structure, and generally raised some issues with clarity, notation, etc. The author produced clear rebuttals and the revised text was judged, by each reviewer, to adequately address their concerns. I am also satisfied with the reviews, rebuttals, and revisions.

**Audience:**

The paper contributes to the literature on causal discovery using interventional data. Causal discovery is an important, growing area of interest within ML.

**Claims And Evidence:**

The claims in the paper are backed up by empirical evidence against a range of baselines and on a range of benchmarks of moderate scale. The work justifies the particular baselines.